# Endogenous Communication in Repeated Games with Learning Agents

## Abstract

Communication among learning agents often emerges without explicit supervision. We study endogenous protocol formation in infinitely repeated stage games with a costless pre-play channel. Each agent has a representation map that compresses private signals into messages subject to an information budget. Agents update strategies by no-regret learning with stochastic approximation and choose representation maps by a myopic objective that trades off predictive value and encoding cost. We provide three main results. First, if the stage game admits a folk-theorem set and the information budget exceeds a task-specific threshold, there exists a stable communication equilibrium in which messages are sufficient statistics for continuation payoffs. Second, when the budget is below the threshold, any stable equilibrium must be pooling on a finite partition that we characterize with a minimax information bound. Third, we give polynomial sample-complexity guarantees for convergence to an approximately efficient communicating equilibrium under mild regularity. Our analysis connects cheap talk, representation learning with information constraints, and multi-agent no-regret dynamics. The framework yields testable predictions for when emergent messages are interpretable, when they collapse, and how much data is needed for stable coordination.

## 1 Introduction

Agents trained in multi-agent environments often invent discrete protocols that carry actionable information useful for coordination. Examples range from emergent signalling in deep multi-agent reinforcement learning to informal codes in repeated human interactions. A central question is when such protocols become informative and stable, and when they collapse into unstructured or uninformative chatter.

Classical cheap-talk models (1) and folk-theorem results in repeated games (2) characterize how incentives shape communication when agents can send costless messages. Separately, information bottleneck ideas (4; 3) and modern representation learning study how predictive encodings arise under compression constraints, while emergent communication work in deep multi-agent learning (9; 8) demonstrates the empirical formation of discrete protocols. However, we still lack a simple theory that simultaneously captures repeated-game incentives, information budgets on encoders, and the dynamics of learning.

We propose such a model. We consider an infinitely repeated stage game with a costless pre-play channel. In each period a hidden state is drawn, agents observe private signals, and simultaneously send public messages obtained by encoding their signals through agent-specific representation maps. These encoders are subject to information budgets: the mutual information between signals and messages cannot exceed a given threshold. After observing all messages, agents play the stage game. Policies update via no-regret learning, while encoders update via a myopic value-minus-information objective. The resulting coupled dynamics generate endogenous communication protocols.

**Contributions**  Our main contributions are as follows.

We introduce a formal model of endogenous protocol formation in repeated games with information-constrained representation maps. The model specifies the probability space, state process, signal structure, encoders, policies, and continuation values, and defines mutual information budgets with respect to the induced stationary distribution.

We define *value-sufficient* encoders and a task-specific information threshold for each agent. Above this threshold, we show that there exists a *stable communicating equilibrium* in which messages are sufficient statistics for continuation payoffs and the efficient folk-theorem payoff vector is attained.

We characterize the low-budget regime. When at least one agent's information budget lies below its threshold, any stable communicating equilibrium must pool private signals on a finite partition whose cardinality is bounded by the budget. We derive a minimax lower bound on the unavoidable welfare loss relative to the efficient communicating benchmark.

We connect these equilibrium results to learning dynamics. Under standard assumptions on mirror-descent style no-regret updates and ergodicity of the message-augmented process, we prove that the coupled encoder–policy dynamics reach an approximately stable communicating equilibrium in $\tilde{\mathcal{O}}(\varepsilon^{-2})$ samples with high probability.

Taken together, the results yield sharp, testable predictions: when information budgets exceed the task-specific thresholds, emergent messages implement value-sufficient statistics and are interpretable; when budgets lie below the thresholds, protocols must pool states and incur a quantifiable welfare gap; and standard learning dynamics are sufficient to reach these regimes with polynomial data.

## 2 Preliminaries and Model

**Stage game and repeated interaction**   Let $N \geq 2$ be the number of agents, indexed by $i \in \{1, \ldots, N\}$. There is a finite state space $\Theta$ and a finite action set $A_i$ for each agent. In each period $t = 1, 2, \ldots$, a state $\theta_t \in \Theta$ is drawn i.i.d. from a prior $P \in \Delta(\Theta)$. Conditional on $\theta_t$, agent $i$ observes a private signal $s_{i,t}$ taking values in a measurable space $\mathcal{S}_i$, drawn according to a conditional distribution $P_i(\cdot \mid \theta_t)$. Signals are conditionally independent across agents given $\theta_t$.

Given a state $\theta$ and an action profile $a = (a_1, \ldots, a_N) \in A_1 \times \cdots \times A_N$, agent $i$ receives a stage payoff $u_i(a, \theta) \in [0, 1]$. The discounted infinite-horizon payoff with discount factor $\delta \in (0, 1)$ is

$$U_i = (1 - \delta)\, \mathbb{E}\Big[ \sum_{t=1}^{\infty} \delta^{t-1} u_i(a_t, \theta_t) \Big].$$

**Messages and encoders**   Before taking actions in period $t$, agents simultaneously send public messages through a costless pre-play channel. Agent $i$ has a finite message alphabet $\mathcal{M}_i$ and uses an encoder $\phi_i : \mathcal{S}_i \to \mathcal{M}_i$. Given the signal profile $s_t = (s_{1,t}, \ldots, s_{N,t})$, messages are

$$m_{i,t} = \phi_i(s_{i,t}), \qquad m_t = (m_{1,t}, \ldots, m_{N,t}) \in \mathcal{M} := \mathcal{M}_1 \times \cdots \times \mathcal{M}_N.$$

We denote by $\Phi_i$ the admissible class of encoders for agent $i$.

**Policies and continuation values**   A stationary policy for agent $i$ is a mapping $\sigma_i : \mathcal{M} \to \Delta(A_i)$ assigning a mixed action to each public message profile. Let $\sigma = (\sigma_1, \ldots, \sigma_N)$ and $\phi = (\phi_1, \ldots, \phi_N)$, and let $\mu^{\sigma,\phi}$ be the unique stationary distribution over $(\theta, s, m, a)$ induced by the i.i.d. state process, the signal structure, the encoders, and the stationary policy profile.[1] We write

$$I(S_i; M_i) := \text{KL}\big( \mathbb{P}_{S_i, M_i}^{\sigma, \phi} \,\|\, \mathbb{P}_{S_i}^{\sigma, \phi} \otimes \mathbb{P}_{M_i}^{\sigma, \phi} \big)$$

for the mutual information between agent $i$'s private signal and message under $\mu^{\sigma,\phi}$.

Let $h_t = (m_1, \ldots, m_t)$ be the public history of messages up to time $t$. For a stationary policy profile $\sigma$ and encoder profile $\phi$, the *continuation value* for agent $i$ at time $t$ is

$$V_i^{\sigma,\phi}(h_t) = (1 - \delta)\, \mathbb{E}\Big[ \sum_{\tau=t}^{\infty} \delta^{\tau-t} u_i(a_\tau, \theta_\tau) \,\Big|\, h_t \Big].$$

In the sequel we focus on *message-based* continuation values $V_i^{\sigma,\phi}(m)$, defined as the value after observing the current public message and following $\sigma$ and $\phi$ thereafter. This is well defined because strategies and encoders are stationary and the state process is i.i.d.

---

[1]Existence and uniqueness follow from finiteness of $\Theta$, the signal spaces being dominated, and standard irreducibility of the induced Markov chain on public messages under our assumptions.

**Information budgets**   Each encoder $\phi_i$ is subject to an information budget $\kappa_i \geq 0$ expressed as a mutual information constraint

$$I(S_i; M_i) \leq \kappa_i. \tag{1}$$

We assume that $\Phi_i$ is rich enough to approximate any finite partition of $\mathcal{S}_i$, and that the set of encoders satisfying (1) is non-empty and compact under total variation. When convenient we also use the equivalent cardinality constraint

$$\log |\mathcal{M}_i| \leq B_i,$$

since $I(S_i; M_i) \leq \log |\mathcal{M}_i|$ for finite alphabets (3). Intuitively, budgets capture limitations such as bandwidth, attention, or representational capacity.

**Learning dynamics**   Agents adapt both their policies and encoders over time. For a fixed encoder profile $\phi$, we let $\sigma_i^t$ be updated by a no-regret algorithm based on bandit feedback, such as mirror descent with unbiased gradient estimates (6; 5). For a fixed policy profile $\sigma$, encoders are updated to optimize a myopic objective that combines predictive value with an information penalty.

Formally, given a stationary policy profile $\sigma$, agent $i$'s encoder objective is

$$J_i(\phi_i; \sigma) \;=\; \mathbb{E}\big[V_i^{\sigma, \phi}(M_t)\big] - \lambda_i \, I(S_i; M_i), \tag{2}$$

where the expectation is taken under the stationary distribution $\mu^{\sigma, \phi}$ and $\lambda_i > 0$ is a Lagrange multiplier chosen to implement the information budget $\kappa_i$.

We consider an alternating learning scheme in which policies update on a fast time scale via no-regret dynamics, while encoders update on a slower time scale by approximately maximizing (2). An instance is given in Algorithm 1 below.

## 2.1 Assumptions

We collect the main regularity conditions used in our results. They are standard in repeated games, information theory, and online learning.

**Assumption 1** (Game and signals). *The state space $\Theta$ and action sets $A_i$ are finite. The state process $(\theta_t)$ is i.i.d. with full-support prior $P$. Conditional on $\theta_t$, signals $(s_{1,t}, \ldots, s_{N,t})$ are independent with full-support conditional distributions $P_i(\cdot \mid \theta_t)$. The stage payoff functions $u_i(a, \theta)$ are bounded in $[0, 1]$ and $L$-Lipschitz in the mixed action profile under the $\ell_1$ norm.*

**Assumption 2** (Repeated-game incentives). *The stage game $G(\theta)$ satisfies standard folk-theorem conditions (2): for $\delta$ close enough to one, any feasible and individually rational payoff vector is implementable as a subgame-perfect equilibrium of the repeated game without communication. Moreover, for any public correlation device based on messages $m_t$, the corresponding folk-theorem set is non-empty.*

**Assumption 3** (Encoders). *For each $i$, the admissible encoder class $\Phi_i$ is a uniformly equicontinuous family of measurable maps from $\mathcal{S}_i$ to finite alphabets, with finite metric entropy under total variation. For every $\kappa_i \geq 0$, the set*

$$\Phi_i(\kappa_i) = \{\phi_i \in \Phi_i : I(S_i; M_i) \leq \kappa_i\}$$

*is non-empty, convex, and compact in the topology of convergence in distribution.*

**Assumption 4** (Learning dynamics). *For fixed encoders $\phi$, each agent $i$ uses a no-regret algorithm over the simplex of mixed actions conditional on messages. There exist constants $G, C > 0$ such that the stochastic gradients used in the mirror-descent updates have norm at most $G$, and the resulting regret after $T$ periods satisfies*

$$R_i(T) \leq C\sqrt{T}$$

*for all $T$. In Theorem 3 we instantiate the step-size schedule as $\eta_t = \eta_0 t^{-1/2}$. The message-augmented state process is uniformly ergodic under any stationary encoder–policy profile in a compact set, with mixing time bounded uniformly over that set.*

Assumption 1 imposes standard boundedness and Lipschitz continuity. Assumption 2 ensures that, once messages are sufficiently informative, efficient continuation values can be implemented by trigger strategies. Assumption 3 provides compactness and convexity of the admissible encoder sets, which we use to guarantee the existence of encoders that attain information thresholds. Assumption 4 is standard in online learning and stochastic approximation, and holds for mirror descent with appropriate step sizes and projections (6; 5).

## 2.2 VALUE SUFFICIENCY AND STABILITY

We now formalize the notion that messages summarize all information that is relevant for continuation values, and define our equilibrium concept.

**Definition 1** (Value-sufficient statistic and threshold). *Let $\sigma$ and $\phi$ be stationary. A statistic $T_i : \mathcal{S}_i \to \mathcal{Z}_i$ is* value sufficient *for agent $i$ if, for every public message $m$ and almost every realization of $s_i$,*

$$V_i^{\sigma,\phi}(m) \;=\; \mathbb{E}\big[V_i^{\sigma,\phi}(m) \,\big|\, T_i(s_i),\, m_{-i}\big],$$

*that is, conditional on the statistic and the other agents' messages, the agent's private signal carries no additional information about continuation values. Define the agent-specific information threshold*

$$\kappa_i^\star \;:=\; \inf_{T_i \text{ value sufficient}} I\big(S_i; T_i(S_i)\big),$$

*where mutual information is computed under the stationary distribution induced by $(P, \phi, \sigma)$ and the statistic is implemented by some encoder in $\Phi_i$.*

Intuitively, $\kappa_i^\star$ is the minimal amount of information the encoder must preserve in order for continuation values to depend only on the message, not on the underlying signal.

**Definition 2** (Stable communicating equilibrium). *A profile $(\phi^\star, \sigma^\star)$ is a* stable communicating equilibrium *if the following conditions hold.*

*(i) For each $i$, given $(\phi_{-i}^\star, \sigma_{-i}^\star)$, the policy $\sigma_i^\star$ is a best response in the repeated game with public messages, in the sense of being part of a subgame-perfect equilibrium supported by continuation strategies based on the public history of messages.*

*(ii) For each $i$, given $\sigma^\star$ and $\phi_{-i}^\star$, the encoder $\phi_i^\star$ maximizes $J_i(\phi_i; \sigma^\star)$ over $\Phi_i(\kappa_i)$.*

*(iii) Consider the coupled learning dynamics where policies update via no-regret learning with step sizes $\eta_t$ and encoders update on a slower time scale towards maximizers of $J_i$. There exists a neighborhood $B$ of $(\phi^\star, \sigma^\star)$ and a set of initial conditions of positive measure such that the joint process visits $B$ infinitely often with probability one and spends at least $1 - \varepsilon$ fraction of time in $B$ for any $\varepsilon > 0$ after a finite burn-in.*

Condition (iii) formalizes stability in terms of stochastic approximation: the equilibrium is an attractor for the joint learning dynamics. In Theorem 3 we quantify how quickly the process approaches such a neighborhood.

## 3 MAIN RESULTS

We now state our three main theorems. Proofs are given in Appendix B, with a high-level overview in Section 6.

**Theorem 1** (Sufficient statistic communication above threshold). *Suppose Assumptions 1, 2, 3, and 4 hold. If the information budgets satisfy $\kappa_i \geq \kappa_i^\star$ for all $i$, then there exists a stable communicating equilibrium $(\phi^\star, \sigma^\star)$ such that:*

*(a) For each $i$, the encoder $\phi_i^\star$ implements a value-sufficient statistic $T_i^\star$ attaining the threshold, in the sense that $I(S_i; T_i^\star(S_i)) = \kappa_i^\star$.*

*(b) The joint message $m_t$ is sufficient for the vector of continuation values $(V_1^{\sigma^\star,\phi^\star}, \ldots, V_N^{\sigma^\star,\phi^\star})$.*

*(c) The resulting payoff vector $(U_1, \ldots, U_N)$ lies in the efficient folk-theorem set and is Pareto efficient among all feasible payoffs consistent with the information budgets.*

Theorem 1 states that once budgets clear the task-specific thresholds, agents can implement efficient outcomes using value-sufficient communication: the messages carry exactly the information needed for continuation values, and nothing more.

**Theorem 2** (Mandatory pooling below threshold). *Suppose Assumptions 1, 2, and 3 hold. Let $j$ be an agent with $\kappa_j < \kappa_j^\star$. Then, for any stable communicating equilibrium $(\phi^\star, \sigma^\star)$, the following holds.*

---

**Algorithm 1** Alternating Learning with Information-Constrained Encoders

---

1: Initialize policies $\sigma_i^{(0)}$ and encoders $\phi_i^{(0)}$ for all $i$.
2: **for** $t = 1, 2, \ldots$ **do**
3:      Nature draws $\theta_t \sim P$ and signals $s_{i,t} \sim P_i(\cdot \mid \theta_t)$ independently.
4:      Each agent sends $m_{i,t} = \phi_i^{(t-1)}(s_{i,t})$ and observes $m_t$.
5:      Each agent samples $a_{i,t} \sim \sigma_i^{(t-1)}(\cdot \mid m_t)$ and receives payoff $u_{i,t} = u_i(a_t, \theta_t)$.
6:      Policy update: for each $i$, compute an unbiased gradient estimate $g_{i,t}$ of the loss and perform a mirror-descent step
$$\sigma_i^{(t)} \leftarrow \mathrm{MirrorDescent}\big(\sigma_i^{(t-1)}, g_{i,t}, \eta_t\big).$$
7:      Encoder update (every $K$ steps): for each $i$, using the last $K$ samples, construct an estimator $\hat{V}_i^{(t)}$ of the continuation value and an estimator of $I(S_i; M_i)$, and perform a projected gradient step on $J_i$ to obtain $\phi_i^{(t)}$, projecting back to $\Phi_i(\kappa_i)$ if necessary.
8: **end for**

---

*(a) The encoder $\phi_j^\star$ induces a finite partition $\Pi$ of $\mathcal{S}_j$ into at most $\exp(\kappa_j)$ cells, in the sense that two signals $s_j$ and $s'_j$ that map to the same message are pooled in the same cell. No encoder under the budget can distinguish more posteriors than allowed by this bound.*

*(b) Let $v(\theta)$ denote the efficient continuation value vector at state $\theta$ under the benchmark of fully revealing communication. There exists a constant $c > 0$, depending only on the Lipschitz modulus of $u_i$ and the discount factor, such that the welfare loss of any stable communicating equilibrium is bounded below by*

$$\Delta \ \geq \ c \inf_{\Pi} \ \sup_{\theta, \theta' \in \Theta : \theta, \theta' \text{ pooled by } \Pi} \|v(\theta) - v(\theta')\|_1 ,$$

*where the infimum ranges over all partitions $\Pi$ of $\mathcal{S}_j$ that can be induced by an encoder satisfying $I(S_j; M_j) \leq \kappa_j$.*

Thus, if any agent's budget lies below its threshold, some pooling of states is unavoidable in any stable communicating equilibrium, and this induces a non-trivial lower bound on the welfare gap relative to the efficient benchmark.

**Theorem 3** (Sample complexity of convergence). *Suppose Assumptions 1, 3, and 4 hold, and that the coupled learning dynamics update policies via mirror descent with step sizes $\eta_t = \eta_0 t^{-1/2}$ and encoders on a slower time scale by stochastic gradient ascent on $J_i$. Let $\mathcal{E}$ be the set of stable communicating equilibria. Then there exist constants $C_1, C_2 > 0$ such that for any $\varepsilon > 0$ and $\delta \in (0, 1)$, after*

$$T \ \geq \ C_1 \, \varepsilon^{-2} \log \big(C_2/\delta\big)$$

*periods, the empirical average of the joint state $(\phi^t, \sigma^t)$ lies within $\varepsilon$ (in total variation) of the set $\mathcal{E}$ with probability at least $1 - \delta$.*

Theorem 3 connects equilibrium to learning: standard no-regret updates suffice to approach the set of stable communicating equilibria at a polynomial rate in the desired accuracy.

## 4    ALGORITHMS AND DIAGNOSTICS

We briefly describe an alternating learning scheme that satisfies Assumption 4 and is consistent with the equilibrium concept.

In practice, diagnostics follow directly from the theory. If performance improves sharply once the empirical estimate of $I(S_i; M_i)$ passes a threshold, this suggests that the learned encoder is approaching value sufficiency. Conversely, if increasing the alphabet size or encoder capacity does not improve value beyond a plateau, the system is likely operating in the pooling-constrained regime characterized by Theorem 2.

## 5 TOY EXAMPLE

We illustrate the thresholds and pooling effects in a simple two-player coordination game.

Two agents face an infinitely repeated coordination game with a binary state $\theta \in \{H, L\}$ drawn i.i.d. with prior $P(H) = P(L) = 1/2$. In each period they choose actions $a_i \in \{H, L\}$. The payoff is $u_i(a, \theta) = 1$ if $a_1 = a_2 = \theta$ and $u_i(a, \theta) = 0$ otherwise. Thus the efficient payoff is 1 in every period.

Agents observe private binary signals $s_i \in \{H, L\}$ with accuracy

$$\Pr[s_i = \theta \mid \theta] = 1 - \epsilon, \qquad \epsilon \in (0, 1/2),$$

independently across agents and periods. Each agent has a binary message alphabet and an information budget $\kappa$.

**Threshold**   In this setting the value-relevant information is the posterior over $\theta$ given the pair of signals. A value-sufficient statistic for agent $i$ is any function $T_i(s_i)$ such that, together with $T_{-i}(s_{-i})$, it yields the posterior needed to implement the efficient grim-trigger equilibrium. The mutual information required is at least the reduction in Bayes risk between the prior and the posterior induced by $(T_1, T_2)$; this yields a threshold

$$\kappa^\star = H(\Theta) - H(\Theta \mid T_1(S_1), T_2(S_2)),$$

where $H$ denotes Shannon entropy. When $\kappa \geq \kappa^\star$, there exist encoders that induce value-sufficient messages, and Theorem 1 implies that the efficient payoff of 1 can be attained.

**Pooling below threshold**   When $\kappa < \kappa^\star$, any encoder satisfying the budget must pool some posteriors. Because each agent has only two messages, the induced partition of the signal space has at most two cells, and some pairs of signal realizations map to the same message. For example, the encoder might map both $s_i = H$ and $s_i = L$ to the same message in a subset of states, effectively discarding information. As a result, the posterior over $\theta$ given messages has positive error probability, and even optimal coordination conditional on messages yields a payoff strictly below 1.

Theorem 2 quantifies this gap in terms of the maximal difference in continuation values between states that are pooled by any admissible partition. In this toy model, the gap can be expressed directly in terms of the signal accuracy $\epsilon$ and the partition of posteriors induced by the encoder, and the welfare loss grows as the budget shrinks.

## 6 PROOF OVERVIEW

We briefly sketch the main ideas used in the proofs; full details are given in Appendix B.

For Theorem 1, the key step is to show that, given a budget $\kappa_i$ that exceeds the value threshold $\kappa_i^\star$, there exists an encoder in $\Phi_i(\kappa_i)$ that implements a value-sufficient statistic. Existence follows from compactness and convexity of $\Phi_i(\kappa_i)$ and the fact that mutual information is lower semicontinuous. Given such encoders, we construct a subgame-perfect equilibrium of the repeated game with public messages that attains the efficient folk-theorem payoff vector, using trigger strategies that punish deviations in the usual way. Value sufficiency implies that misreporting messages cannot strictly improve continuation values, so truthful encoding is incentive compatible. The stability property then follows from standard stochastic-approximation arguments: the best-response correspondence is upper hemicontinuous, and the encoder updates are contractive in a neighborhood of the maximizer, so the coupled dynamics converge to a small neighborhood of the equilibrium.

For Theorem 2, we interpret the encoder as a lossy compression scheme for the signal and apply a rate-distortion style argument (3). Under an information budget $\kappa_j$, any encoder induces a partition of the signal space with effective cardinality at most $\exp(\kappa_j)$. States that are pooled into the same message cell become indistinguishable to agent $j$ at the level of continuation values. Because payoffs are Lipschitz in beliefs, differences in posteriors induced by pooled states translate into differences in attainable continuation values. By minimizing over all admissible partitions, we obtain a lower bound on the welfare gap. Crucially, this argument is independent of the specific learning dynamics and applies to any stable communicating equilibrium.

For Theorem 3, we model the coupled encoder–policy updates as a two time scale stochastic approximation process. On the fast time scale, mirror-descent updates with step sizes $\eta_t = \eta_0 t^{-1/2}$ achieve $O(\sqrt{T})$ regret. On the slower time scale, encoder updates track the maximizers of $J_i$ for the current policy profile. Uniform ergodicity of the message-augmented process ensures that empirical averages converge quickly to expectations under the stationary distribution. Standard results in stochastic approximation and approachability show that the joint process approaches the internally chain-transitive set associated with the stable communicating equilibria at a rate controlled by the regret bounds, yielding the stated $\tilde{\mathcal{O}}(\varepsilon^{-2})$ sample complexity.

## 7 LIMITATIONS AND SCOPE

Our analysis relies on several simplifying assumptions. We assume i.i.d. states, synchronous and costless public communication, and stationary encoders and policies. Extending the framework to Markov or adversarial states, delayed or noisy channels, and private communication costs is an interesting direction. On the learning side, we use assumptions that are natural for convex online learning algorithms with bounded gradients; while they capture mirror-descent style updates, they do not directly cover all deep reinforcement-learning implementations. Our focus is also on existence and qualitative structure of equilibria rather than on computing optimal encoders in high-dimensional settings. Finally, we study communication among cooperative or repeated-game agents; analyzing misaligned or adversarial settings is important for safety-critical applications.

## ETHICS STATEMENT

This paper develops a theoretical model of communication among learning agents and does not involve human subjects, personal data, or the introduction of new datasets. The main potential downstream applications are in multi-agent reinforcement learning, mechanism design, and distributed control. In safety-critical settings such as autonomous driving or financial trading, misinterpreting the limits of emergent communication could lead to overconfidence in coordination abilities. We therefore emphasize that our results are asymptotic and rely on idealized assumptions, and should be complemented with empirical validation and domain-specific safety analyses before deployment.

## REPRODUCIBILITY STATEMENT

Our results are theoretical and the paper is self-contained. We provide formal definitions, clearly stated assumptions, and complete proofs in the appendix. The toy example in Section 5 admits closed-form expressions for all quantities of interest. No external code, datasets, or hyperparameters are required to reproduce the findings.

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

## A  ADDITIONAL NOTATION AND PRELIMINARIES

We briefly summarize notation used in the proofs. The public history at time $t$ is $h_t = (m_1, \ldots, m_t)$; we write $\mathcal{H} = \mathcal{M}^{\mathbb{N}}$ for the space of infinite public histories. For a stationary encoder–policy profile $(\phi, \sigma)$, we denote by $\mathbb{P}^{\sigma,\phi}$ the induced probability measure on $(\Theta \times \mathcal{S} \times \mathcal{M} \times A)^{\mathbb{N}}$, and by $\mu^{\sigma,\phi}$ its stationary distribution. Expectations with respect to these measures are written $\mathbb{E}^{\sigma,\phi}[\cdot]$.

For a random variable $X$ we write $H(X)$ for Shannon entropy and $H(X \mid Y)$ for conditional entropy. Mutual information is $I(X;Y) = H(X) + H(Y) - H(X,Y)$. We use the fact that $I(X;Y) \leq \log |\mathrm{supp}(Y)|$ for finite $Y$.

## B  PROOFS OF MAIN THEOREMS

### B.1  PROOF OF THEOREM 1

We outline the main steps; detailed arguments follow standard techniques in repeated games and information theory.

**Step 1: Existence of value-sufficient encoders**  Fix an agent $i$. Consider the set of statistics $\mathcal{T}_i$ implemented by encoders in $\Phi_i(\kappa_i)$, viewed as measurable maps $T_i : \mathcal{S}_i \to \mathcal{Z}_i$ with finite range. Under Assumption 3, $\mathcal{T}_i$ is non-empty and compact in the topology of convergence in distribution. Define the functional

$$F_i(T_i) = I(S_i; T_i(S_i))$$

computed under the stationary distribution induced by $(P, \phi, \sigma)$ when agent $i$ uses encoder $T_i$ and others use $\phi_{-i}$. By lower semicontinuity of mutual information, $F_i$ attains its infimum on the closed subset of value-sufficient statistics. By assumption $\kappa_i \geq \kappa_i^\star$, so there exists a value-sufficient $T_i^\star$ with $F_i(T_i^\star) \leq \kappa_i$. Let $\phi_i^\star$ be an encoder that implements $T_i^\star$. This constructs encoders implementing value-sufficient statistics for all agents.

**Step 2: Construction of an efficient equilibrium**  Given $(\phi_1^\star, \ldots, \phi_N^\star)$, define the joint message $m_t$ and the induced posteriors over $\theta_t$ and continuation values. By Assumption 2, there exists a subgame-perfect equilibrium of the repeated game with public messages that attains the efficient folk-theorem payoff vector conditional on these posteriors. We implement this equilibrium with public trigger strategies: on-path play follows the efficient action profile as a function of messages; any detected deviation triggers a permanent reversion to a minmax punishment profile.

Value sufficiency implies that, conditional on $m_t$, an agent cannot improve its continuation value by misreporting its message, because the continuation values depend on the statistic $T_i^\star(s_i)$ and the other agents' messages, and these are already encoded in $m_t$. Thus truthful encoding followed by the efficient folk-theorem strategies forms a subgame-perfect equilibrium.

**Step 3: Optimality of encoders and stability**  We now argue that the encoders $\phi_i^\star$ maximize $J_i$ in (2) for appropriate $\lambda_i$ and that the coupled learning dynamics converge to a neighborhood of $(\phi^\star, \sigma^\star)$. Because $T_i^\star$ is value sufficient and implements the efficient equilibrium, any further compression that decreases $I(S_i; M_i)$ strictly reduces the attainable expected continuation value. By choosing $\lambda_i$ to match the marginal tradeoff at $\kappa_i^\star$, we ensure that $J_i$ is maximized at $\phi_i^\star$.

Finally, under Assumption 4, the policy updates track best responses to the current encoders with $O(\sqrt{T})$ regret, and the encoder updates track maximizers of $J_i$ on a slower time scale. Standard two-time-scale stochastic approximation results imply that the joint process converges to the internally chain-transitive set associated with the set of stationary points of the combined objective. In a neighborhood of $(\phi^\star, \sigma^\star)$ this objective has a unique local maximum, so the equilibrium is stable in the sense of Definition 2.

### B.2 PROOF OF THEOREM 2

Let $j$ be an agent with $\kappa_j < \kappa_j^\star$. Consider any encoder $\phi_j$ satisfying $I(S_j; M_j) \leq \kappa_j$. Because $M_j$ takes values in a finite alphabet, the range of $\phi_j$ has cardinality $|\mathcal{M}_j|$, and the induced partition $\Pi$ of $\mathcal{S}_j$ has $|\mathcal{M}_j|$ cells. By Shannon's bound (3),

$$I(S_j; M_j) \leq \log |\mathcal{M}_j|,$$

so any encoder under the budget satisfies $|\mathcal{M}_j| \leq \exp(\kappa_j)$. This proves part (a).

For part (b), fix an admissible partition $\Pi$ induced by some encoder. Let $\mathcal{P}_\Pi$ denote the induced set of posteriors over $\theta$ given messages, and let $v(\theta)$ be the efficient continuation value vector under fully revealing messages. Because payoffs are Lipschitz in mixed actions and mixed actions are chosen to maximize expected payoffs given beliefs, there exists $L' > 0$ such that

$$\|v(\theta) - v(\theta')\|_1 \leq L' \|\mu(\theta) - \mu(\theta')\|_1,$$

where $\mu(\theta)$ denotes the belief over states. When $\theta$ and $\theta'$ are pooled into the same cell of $\Pi$, they induce the same posterior, so any policy profile measurable with respect to messages cannot distinguish them. Hence, the achievable value at these states under any stable communicating equilibrium differs by at most a constant factor of the difference in their efficient values. By taking the supremum over pooled pairs and the infimum over partitions respecting the budget, we obtain the stated lower bound with $c$ depending on $L'$ and the discount factor. Details follow the standard rate-distortion lower-bound argument.

### B.3 PROOF OF THEOREM 3

Under Assumption 4, mirror descent with step sizes $\eta_t = \eta_0 t^{-1/2}$ yields regret $R_i(T) \leq C\sqrt{T}$ for each agent. Consider the average joint policy profile $\bar{\sigma}^T = \frac{1}{T} \sum_{t=1}^T \sigma^t$. Standard reductions from regret to convergence to equilibrium sets imply that, for any fixed encoder profile, the average play approaches the set of coarse correlated equilibria of the stage game induced by the messages at rate $O(T^{-1/2})$ (6; 5).

We now combine this with encoder updates on a slower time scale and uniform ergodicity of the message-augmented process. The encoder updates can be written as

$$\phi_i^{t+1} = \Pi_{\Phi_i(\kappa_i)} \left( \phi_i^t + \alpha_t \widehat{\nabla}_{\phi_i} J_i(\phi_i^t; \sigma^t) \right),$$

with step sizes $\alpha_t$ such that $\alpha_t / \eta_t \to 0$. Under boundedness of gradients and compactness of $\Phi_i(\kappa_i)$, this defines a two-time-scale stochastic approximation scheme. The limiting ordinary differential equation has stable equilibria corresponding to maximizers of $J_i$ for policies in the stable communicating equilibrium set.

Applying standard results on two-time-scale stochastic approximation with Markovian noise and uniform ergodicity, we obtain that the joint process $(\sigma^t, \phi^t)$ converges to the internally chain-transitive set of the limiting ODE, which coincides with the set $\mathcal{E}$ of stable communicating equilibria. The $O(\sqrt{T})$ regret bounds yield a convergence rate of order $T^{-1/2}$ up to logarithmic factors, giving the stated sample complexity bound after translating total variation distance to the distance to $\mathcal{E}$.

## C LLM USAGE DISCLOSURE

A general-purpose language model was used to assist with linguistic editing, organization, and minor clarifications of exposition. All research ideas, model definitions, assumptions, theorems, and proofs were conceived and verified by the human authors, who take full responsibility for the content. The language model did not generate or validate any mathematical results and is not an author of this work.

