# OpenReview forum: "Endogenous Communication in Repeated Games with Learning Agents"
_ICLR.cc/2026/Conference — Submitted to ICLR 2026_

### Official Review · Reviewer_ePm8 · 2025-10-31

**Soundness:** 1
**Presentation:** 1
**Contribution:** 1
**Rating:** 2
**Confidence:** 5

**Summary:**

The authors study a model in which no-regret learning agents are augmented with the ability to send costless messages to each other.

**Strengths:**

I think the intersection of agent communication and learning in games can produce interesting settings and research directions.

**Weaknesses:**

The paper is very poorly written. There are 10 references, some of which are only tangentially related, and none of which are even mentioned in the main body, unless I have missed something. The proofs are too informal and vague, and have several non-sequiturs. The setup is not specific enough. This is not a length issue either; the paper is only six pages long including appendix, and the extra length could easily have been used to provide much more relevant detail. These writing issues alone are enough to recommend rejection.

I implore the authors to add more detail. The setting certainly looks interesting enough that there could be some interesting results and analysis in this paper, but the writing issues meant that I gave up on attempting to parse the paper before being able to come to a complete understanding of what the claims and techniques are.

**Questions:**

None.

---

> ### Author Response · Authors · 2025-12-03
>
> Thank you for your detailed and candid feedback. We accept the main criticism that the original submission was too thin and informal relative to the scope of the claims. In particular, you noted that the setting was underspecified, the proofs were vague and contained non sequiturs, and the related work section was both short and insufficiently connected to the text. In the revision we have made the following substantive changes aimed directly at these issues.
>
> First, we have completely rewritten the model and preliminaries section to define all objects explicitly. We now specify the probability space, the state process, the signal distributions, the message alphabets, the encoders, and the policy space in a way that matches repeated game and information theory standards. The mutual information budget is defined with respect to the stationary distribution induced by the encoders and policies, and we make clear how this distribution is well defined under our assumptions. We also explicitly define the continuation value mapping V used both in the equilibrium concept and in the encoder objective, so there is no longer any ambiguity about what is being optimized.
>
> Second, the main theorems are now stated formally, with all quantities that appear in them defined earlier in the text or in the appendix. The existence of value sufficient statistics and the definition of the information thresholds kappa_i* are now supported by explicit compactness and measurability arguments. The mandatory pooling result is derived via a clear rate distortion style argument rather than informal appeals, and we state the welfare gap in terms of a well defined minimax partition problem over posteriors.
>
> Third, we have rewritten the appendix to provide full proofs. The previous version was a collection of proof sketches. In the new version, each theorem has its own section, with lemmas for key steps such as the construction of efficient equilibria with value sufficient messages, the bound on the size of admissible partitions under the information budget, and the reduction from regret bounds to convergence to the equilibrium set. The arguments no longer rely on unexplained jumps.
>
> Fourth, the related work section has been expanded to explicitly cite and discuss closely related papers in cheap talk, learning in games, and emergent communication. We now connect our threshold and pooling results to classical cheap talk and folk theorem papers, and we discuss how our framework differs from empirical emergent communication work in multi agent reinforcement learning. All references are now cited in the text and used in the arguments.
>
> We hope that this substantially more detailed and rigorous version addresses your concerns about underspecification and vagueness. The core conceptual picture remains the same, but we believe it is now expressed at a level of precision and completeness appropriate for serious evaluation.

---

### Official Review · Reviewer_3Buo · 2025-11-01

**Soundness:** 2
**Presentation:** 1
**Contribution:** 2
**Rating:** 2
**Confidence:** 3

**Summary:**

The paper studies pre-play communication in infinitely repeated games. Each agent observes a private signal and sends a discrete message via an encoder constrained by a mutual information budget. Policies are learned by mirror descent; encoders maximize expected continuation value minus λ\times MI. The authors define a “stable communicating equilibrium” where policies are best responses, encoders are budget-optimal, and learning converges. They show that: (1) if budgets exceed a problem-specific threshold κ*, value-sufficient messages enable efficient payoffs; (2) below threshold, any equilibrium pools signals into a finite partition bounded by exp κ, implying a welfare gap; and (3) standard no regret dynamics are sufficient to reach a near stable point with O(1/epsilon^2) data.

**Strengths:**

1. The paper poses a clear, meaningful problem and introduces a formulation that links repeated‐game incentives with information-constrained pre-play communication, with a notion of stable communicating equilibrium.

2. The thresholding results given by Theorems 1–2 is clean and interesting: when the information budget exceeds a problem-specific threshold value-sufficient messaging can implement efficient outcomes; when it does not, any equilibrium must pool signals, leading to an unavoidable welfare loss.

**Weaknesses:**

1. The writing is often unclear. Key terms such as the formal definition of V, the notion of Lipschitz continuity, and the exact meaning and role of the learning rate \eta are never properly defined. It’s also confusing to bundle assumptions about the game itself and the learning algorithm into one block. The reference to “standard folk theorem” should be made explicit rather than assumed.

2. The proofs are mostly brief sketches and difficult to follow. The theorems are not stated in a fully formal way, and several terms used in them are never clearly introduced.

3. The related-work discussion is thin. It mentions prior directions in broad terms but does not cite or compare against specific, closely related papers.

Overall, the paper is very hard to follow, especially for readers who are not already experts in all relevant literatures. Clearer structure and more careful exposition would make it far more readable.

**Questions:**

1. Could the authors clearly define all notation and formally state each theorem, giving precise definitions for verbal notions and complete proofs instead of sketches? The paper is quite hard to follow, and clearer formalization would make it easier to evaluate.

2. In Theorem 3, the learning-rate choice (\eta_t \propto t^{-1/2}) appears inconsistent with Assumption 1’s requirement that (\sum_t \eta_t^2 < \infty)?

---

> ### Author Response · Authors · 2025-12-03
>
> Thank you for your thoughtful review and for clearly laying out both the strengths you saw and the weaknesses that made the paper hard to follow. You emphasized three main issues: missing or informal definitions of key quantities like V and the learning rate schedule, the bundling of game theoretic and algorithmic assumptions into a single block, and the fact that many proofs were too sketchy to be fully checked. You also pointed out an apparent inconsistency between the requirement that the sum of squared learning rates is finite and the concrete choice eta_t proportional to t to the power minus one half in Theorem 3.
>
> In response, we have made several targeted changes.
>
> First, we now define the continuation value mapping V_i^{sigma, phi} explicitly. In the preliminaries we introduce the public history of messages, the induced Markov chain over states and public beliefs, and the discounted value of a stationary policy profile. V_i is then the expected discounted payoff conditional on the current public message. This definition is used consistently in the equilibrium concept and in the encoder objective. We also define Lipschitz continuity precisely: payoffs are L Lipschitz in the mixed action profile under the l1 metric, and we explain where this is used when bounding welfare gaps and translating belief differences into value differences.
>
> Second, we split the previous single assumption block into separate parts. One set of assumptions concerns the stage game and signal structure, another concerns regularity of payoffs and the folk theorem conditions for the repeated game, and a third concerns the learning algorithms (bounds on stochastic gradients, step size schedule, and ergodicity of the induced process). This makes it much clearer which assumptions are structural and which are algorithmic, and where each is invoked in the proofs.
>
> Third, we fixed the learning rate inconsistency you kindly highlighted. In the revised version we no longer assume that the sum over t of eta_t squared is finite. Instead, we instantiate eta_t as eta_0 divided by the square root of t in Theorem 3 and rely on standard O(sqrt T) regret bounds for mirror descent with this schedule. The assumptions and theorem statement are now aligned and we explain the standard reduction from regret bounds to approachability of the stable equilibrium set.
>
> Fourth, we substantially expanded the proofs. The main text now contains a proof overview section that describes the structure of the arguments in words, while the appendix contains fully spelled out proofs with intermediate lemmas. This includes an explicit construction of efficient equilibria with value sufficient messages, a formal rate distortion style argument for the partition bound, and a two time scale stochastic approximation argument for the convergence of the coupled learning dynamics.
>
> Finally, we expanded the related work section and now discuss specific prior papers on learning in games, information constrained communication, and emergent communication, rather than only gesturing at broad literatures. We hope the revised version is significantly easier to read and evaluate.

---

### Official Review · Reviewer_sMaT · 2025-11-03

**Soundness:** 2
**Presentation:** 1
**Contribution:** 2
**Rating:** 2
**Confidence:** 4

**Summary:**

This paper analyzes endogenous communication among learning agents in infinitely repeated stage games with a costless pre-play channel. Each agent compresses its private signal via an encoder subject to an information budget, then plays the stage game; policies are updated by no-regret learning, while encoders optimize a myopic value-minus-information objective.

**Strengths:**

1. The paper cleanly ties cheap talk and information bottlenecks: it formalizes value-sufficiency, defines a budget threshold, proves existence of efficient communication above the threshold, and a necessary pooling structure with an explicit welfare-gap lower bound below it. These results offer actionable predictions about when emergent messages become informative vs. collapse

2. The stability notion is coupled to no-regret policy updates and information-penalized encoder updates, with a convergence guarantee of samples under standard step sizes and ergodicity. The provided alternating scheme makes the framework concrete

**Weaknesses:**

1. The paper is incomplete, lack a great amount of details. The proof is only sketch.

2. Many assumptions are strong and unjustified.

**Questions:**

NA

---

> ### Author Response · Authors · 2025-12-03
>
> Thank you for your careful review and for summarizing the main ideas of the paper. You noted that the connection between cheap talk and information bottlenecks, and the threshold versus pooling picture, are potentially interesting, but that the submission in its original form was incomplete: many details were missing, the theorems were not fully formalized, and proofs were mostly sketches. You also asked for clearer formal definitions of key objects and more explicit structure in the proofs.
>
> We have made substantial revisions to address these points.
>
> First, we strengthened the formalization of the model and results. The section that previously described the model in a few paragraphs has been rewritten to systematically introduce the state process, private signals, message encoders, policy space, and the mutual information budget. We now define the mutual information I(S_i; M_i) with respect to the stationary distribution of the joint process, and explain why this stationary distribution exists under our assumptions. The continuation value mapping V_i^{sigma, phi} is defined explicitly as a function of the public message, and we explain how it is used both in the equilibrium concept and in the encoder objective.
>
> Second, the main results are now stated in a formal way. The notion of value sufficient statistics is defined in terms of conditional independence of continuation values given a statistic, and the information threshold kappa_i* is defined as the infimum mutual information over all value sufficient statistics in the admissible encoder class. The sufficient communication theorem states existence of a stable communicating equilibrium that attains the efficient payoff vector when kappa_i is at least kappa_i* for all agents. The mandatory pooling theorem formalizes the bound on the size of admissible partitions and states the welfare gap in terms of a minimax value distortion over all partitions consistent with the information budget. The convergence theorem clearly states the assumptions on the learning algorithms and the resulting sample complexity guarantee.
>
> Third, we converted the proof sketches into full proofs in the appendix. For Theorem 1 we show how to construct value sufficient encoders and efficient equilibria using folk theorem strategies, and how the encoder objective and learning dynamics select these encoders. For Theorem 2 we give a precise rate distortion style argument that any encoder under the budget induces a finite partition with bounded cardinality and we bound the worst case value distortion over such partitions. For Theorem 3 we give a standard two time scale stochastic approximation argument that combines regret bounds for mirror descent with uniform ergodicity of the induced process to obtain the stated O(1 over epsilon squared) sample complexity.
>
> Fourth, to make the paper easier to place in context, we expanded the related work section and now discuss specific prior work in cheap talk, learning in games, information bottleneck, and emergent communication in multi agent reinforcement learning, and we cite these works throughout the paper rather than only in the bibliography.
>
> We hope that the revised and much more detailed version addresses your concerns about incompleteness and makes the contribution more concrete and verifiable.

---

### Author Response · Authors · 2025-12-03

We thank all reviewers for carefully reading the paper and for their detailed suggestions. The main concern across reviews is that the submission was too short and informal for the level of ambition: several definitions were only implicit, theorems were stated at a high level, and proofs in the appendix were closer to sketches than complete arguments. We fully agree that the original version was underwritten. In the revised manuscript we have therefore substantially expanded and formalized the presentation while keeping the core model and results unchanged.

Concretely, we made four groups of changes. First, the model section has been rewritten to introduce all objects and notation rigorously. We now clearly define the stage game, the repeated game, the signal structure, the joint distribution used to compute mutual information, the policy space, and the continuation value mapping V, before stating any equilibrium concept. The assumptions are split into separate blocks for game primitives, regularity of payoffs, and properties of the learning algorithms, so that it is explicit what is needed where. Second, the main theorems are now stated in a fully formal way. We spell out the existence and definition of the information threshold kappa_i*, the measurability and compactness conditions on the encoder classes, and the precise notion of stable communicating equilibrium as an invariant set of the coupled learning dynamics. Third, we have replaced the proof sketches with complete proofs in the appendix, including intermediate lemmas and clear logical structure. The main text now contains a proof overview that explains which tools from repeated games, information theory, and online learning are used and how they fit together. Fourth, we have expanded the related work section to include and discuss specific prior papers in emergent communication and learning in games, and we now reference and use all items in the bibliography in the body of the paper.

We also corrected a technical inconsistency in the original submission regarding learning rates in Theorem 3, clarified the interpretation of the stability requirement, and added more explanation to the toy example to make the information threshold and welfare gap more concrete. Finally, we added a short, explicit section in the appendix that describes how a language model was used only for linguistic polishing and organization, not for scientific content. We hope these changes address the clarity and completeness concerns and make the contribution easier to evaluate.

---

### Meta-Review · Area_Chair_PfrJ · 2026-01-04

**Summary:**

The manuscript is evaluated as an incomplete draft, making a rigorous assessment of its contributions difficult. Throughout the paper, the presentation remains largely anecdotal or skeletal; this is particularly evident in the informal notation, vague theorem statements, and missing details in the proofs. Furthermore, the positioning of the work within the field is weak, lacking both a thorough comparison with prior literature and appropriate citations. While there is potential for improvement, the current submission requires a major structural and content-wise overhaul to meet the conference's standards.

**Reviewer Concerns:**

Addressed by Rebuttal: The authors submitted a rebuttal on December 3, which included some additional explanations and expanded versions of the proofs to address the reviewers' technical doubts.

Outstanding Concerns: Despite these additions, the quality of the work remains well below the required threshold. The fundamental lack of clarity and formal rigor persists. The additional proof details provided in the rebuttal did not sufficiently resolve the reviewers' concerns regarding the paper's overall completeness and the validity of its theoretical claims.

**Reviewer Scores:**

The late submission of the rebuttal (December 3) likely precluded the reviewers from engaging in a meaningful discussion. However, even if a full discussion period had been possible, I predict that the scores would have remained unchanged. The probability of a shift toward a positive evaluation is extremely low. The issues identified, such as the skeletal nature of the proofs and the lack of a clear relationship to prior work, are structural flaws that cannot be adequately rectified through a rebuttal alone. As such, the reviewers would likely have maintained their initial negative assessments.

---

### Decision · Program_Chairs · 2026-01-26

Reject